# Inflammatory patterns in fixed airflow obstruction are dependent on the presence of asthma

Ida Mogensen[1,2]*, Tiago Jacinto[3], Kjell Alving[4], João A. Fonseca[3], Christer Janson[2], Andrei Malinovschi[1]

1 Dept. of Medical Sciences: Clinical Physiology, Uppsala University, Uppsala, Sweden, 2 Dept. of Medical Sciences: Respiratory, Allergy and Sleep Research, Uppsala University, Uppsala, Sweden, 3 CINTESIS, Faculdade de Medicina da Universidade do Porto & Instituto e Hospital CUF, Porto, Portugal, 4 Pediatric Research, Dept. of Women's and Children's Health, Uppsala University, Uppsala, Sweden

* ida.mogensen@medsci.uu.se

**Data Availability Statement:** The data is available at https://wwwn.cdc.gov/nchs/nhanes/Default.aspx and a complete list of datasets used are available in the Supporting Information files.

## Abstract

### Rationale

Fixed airflow obstruction (FAO) can complicate asthma. Inflammation is a proposed underlying mechanism.

### Objective

Our aim in this cross-sectional investigation was to evaluate the blood leucocyte pattern and level of exhaled nitric oxide in asthmatics and non-asthmatics with or without FAO.

### Methods

A total of 11,579 individuals aged $\geq$20 years from the US National Health and Nutrition Examination Survey were included. They were grouped as: controls without asthma and FAO (n = 9,935), asthmatics without FAO (n = 674), asthmatics with FAO (n = 180) and non-asthmatics with FAO (n = 790). FAO was defined as post-bronchodilator FEV1/FVC < lower limit of normal. Exhaled nitric oxide $\geq$ 25ppb, blood eosinophil levels $\geq$300 cells/μL, and blood neutrophil levels $\geq$5100 cells/μL were defined as elevated. Stratified analyses for smoking and smoking history were performed.

### Results

Elevated blood eosinophil levels were more common in all groups compared to the controls, with the highest prevalence in the group with asthma and fixed airflow obstruction (p<0.01). In a multiple logistic regression model adjusted for potential confounders including smoking, the asthma groups had significantly higher odds ratios for elevated B-Eos levels compared to the control group (odds ratio 1.4, (confidence interval: 1.1–1.7) for the asthma group without fixed airflow obstruction and 2.5 (1.4–4.2) for the asthma group with fixed airflow obstruction). The group with fixed airflow obstruction without asthma had higher odds ratio

**Funding:** The U4 Network (Uppsala University), financial support and salary for Ida Mogensen (IM) Uppsala City Council, financial support Bror Hjerpstedts Stiftelse, financial support The Swedish Heart-Lung Foundation, travel grants (IM) The funders had no role in study design, data collection and analysis, decision to publish, or preparation of the manuscript.

**Competing interests:** The authors have declared that no competing interests exist.

for elevated blood neutrophil levels compared to the controls: 1.4 (1.1–1.8). Smoking and a history of smoking were associated to elevated B-Neu levels.

## Conclusion

Fixed airflow obstruction in asthma was associated with elevated blood eosinophil levels, whereas fixed airflow obstruction without asthma was associated with elevated blood neutrophil levels.

## Introduction

Fixed airflow obstruction (FAO) is a non-reversible condition where the airflow during a forced expiratory maneuver is impaired, manifesting spirometrically as a decreased post-bronchodilatory ratio between forced expiratory volume during the first second ($FEV_1$) and forced vital capacity (FVC). This can be due to airway remodeling associated to asthma [1, 2], or as seen in chronic obstructive pulmonary disease (COPD), damaged airways caused by exposure to toxins such as cigarette smoke leading to structural changes [3]. A proposed pathologic mechanism in the development of FAO is a low-grade inflammation. Increased eosinophils in both blood (B-Eos) and sputum has been associated with FAO or lower lung function in both asthmatics and non-asthmatics [4–8], though results have been contradictory [9–11]. Cigarette smoking has been reported to be related to an increase of leucocytes in blood and sputum in both asthmatics and non-asthmatics [12, 13]. Elevated blood neutrophil (B-Neu) count has also been associated with lower lung function both among asthmatics [14] and in the general population [15]. The fraction of exhaled nitric oxide (FeNO) is mainly produced by inducible NO synthase in the airway epithelium. FeNO seems to be more related to respiratory symptoms [16] than to FAO [17].

The aim of this cross-sectional analysis was to investigate how the inflammatory pattern varied with FAO among asthmatics and non-asthmatics compared with in healthy controls; and how this was influenced by smoking. Our hypothesis was that FAO would be related to more inflammation, measured as higher levels of FeNO, B-Eos and/or B-Neu, but that the inflammatory pattern would differ depending on whether or not asthma was present, indicating different pathophysiological mechanisms behind FAO.

## Materials and methods

The participants were collected from the US National Health and Nutrition Examination Surveys (NHANES) 2007–08, 2009–10 and 2011–12, a population based survey [18]. After exclusion of subjects under 20 years of age and subjects who did not have spirometry data or where the spirometries did not fulfill the American Thoracic Society (ATS) criteria regarding acceptability and reproducibility of measurements [19], the population consisted of 11,579 individuals (Fig 1). Subjects were divided into four groups: a reference group (controls) without asthma and a $FEV_1$/FVC ratio above the lower limit of normal (LLN) pre- or post-bronchodilation (defined as without FAO), a group with asthma without FAO, a group with asthma with FAO, and a group with FAO without asthma. These groups were compared regarding the levels of the inflammatory markers.

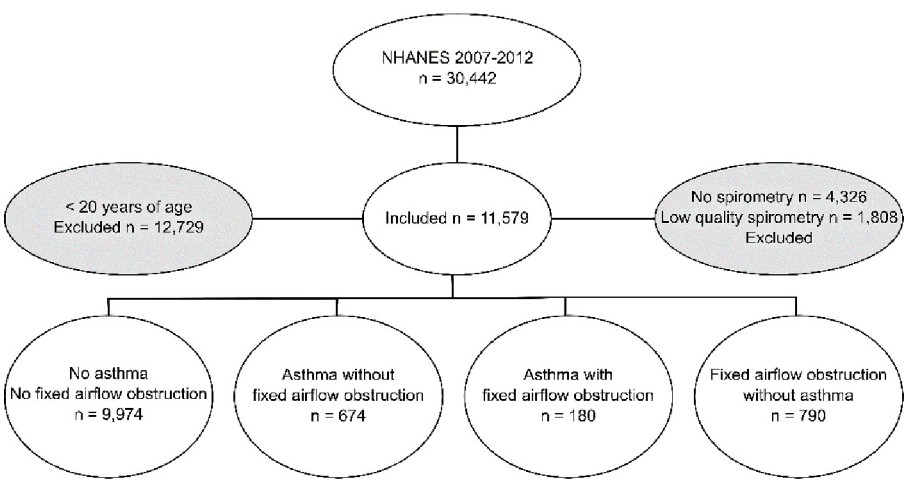

**Fig 1. Flow-chart of included individuals.**

## Spirometry, questionnaires and body measures

$FEV_1$ and FVC were measured with an Ohio 822/827 dry-rolling seal volume spirometer. For the subjects with a non-obstructive pattern ($FEV_1/FVC \geq 0.7$ and above LLN), only pre-bronchodilatory values were collected. However, for subjects with an obstructive pattern ($FEV_1/FVC < 0.7$ or below LLN [20, 21]), bronchodilation was performed with two puffs of albuterol, followed by post-bronchodilation spirometry. The Hankinson reference values were used and corrected for ethnicity [20, 21]. FAO was regarded as present if the $FEV_1/FVC$ ratio was below LLN post-bronchodilation.

Questionnaire-reported variables (S1 Questionnaires) were obtained through interviews performed by trained interviewers [22]. Hay fever was defined as present if the participant reported an episode of hay fever in the preceding year. A smoking history of less than 100 cigarettes defined never smoking. Participant with a smoking history in excess of that were divided into current smokers and ex-smokers. Smoking history (pack-years) was based on the amount of smoking reported. Ethnicity was defined as Mexican-American, other Hispanic, non-Hispanic white, non-Hispanic black, or other ethnicity, including multi-ethnic and Asian. Asthma diagnosis was considered present if the participant reported current asthma diagnosed by a health professional. Asthma morbidity was self-reported, defined as $\geq 2$ different asthma symptoms reported in the preceding year [22]. This has been described in detail previously [23]. Information about medication prescribed for wheezing in the preceding 12 months, and for asthma in the preceding 3 months, and the use of inhaled corticosteroids, combinations of inhaled corticosteroids and long-acting beta-2 agonists, oral corticosteroids, and leukotriene-receptor antagonists in the preceding month was specified. Use of oral or inhaled corticosteroids during the two days preceding FeNO measurement was the subject of a specific question.

Body mass index (BMI) was calculated by dividing weight in kilograms by the square of length in meters. The participants divided into groups: BMI < 25, 25–30 and > 30. A serum cotinine $\geq 3$ ng/mL was regarded as a marker for cigarette smoke exposure [24].

## Biomarkers

FeNO was measured using a handheld device with an electrochemical sensor (NIOX MINO; Aerocrine, Stockholm, Sweden) at an exhalation rate of 50 mL/s. The mean was calculated

after two reproducible measurements and used as the participant's FeNO value. A value $\geq$ 25 ppb was defined as elevated in accordance with the ATS guidelines [25].

B-Eos and B-Neu were measured in venous blood using a Beckman Coulter HMX (Beckman Coulter, Fullerton, CA, USA). Both B-Eos and B-Neu were reported with a resolution of 100 cells/µL. B-Eos counts $\geq$ 300 cells/µL [26, 27] and B-Neu counts $\geq$ 5,100 cells/µL (upper quartile) were defined as elevated, respectively.

## Statistical analyses

The statistical analyses were performed in Stata/IC 15 (Stata Corp, College Station, TX, USA). The three groups, asthma without FAO, asthma with FAO and FAO without asthma, were all compared with the control group regarding baseline characteristics, and levels of FeNO, B-Eos, and B-Neu. Normally distributed continuous variables were analyzed with Wald test and categorical variables with Pearson's chi-squared tests. Non-normally distributed variables were analyzed non-parametrically with Wilcoxon rank sum test. Analyses were stratified for smoking status (never, ex- and current) and smoking history (dichotomous variable, using 10 pack-years as cut-off) and the prevalence of elevated markers was analyzed in between the case-groups. The asthma characteristics were compared between asthmatics with and without FAO.

Adjusted multiple logistic regression models with the outcomes elevated levels of FeNO, B-Eos, and B-Neu, adjusted for sex, ethnicity, age, steroid use in the preceding two days, study year, cotinine levels, use of anti-inflammatory medications, and smoking status were used. Interaction analyses in adjusted logistic models were done for sex and BMI respectively to evaluate effect modification by these variables.

The analyses took into account the complex multistage sampling and sampling weights provided by the NHANES [28]. A p value $\leq$ 0.05 was considered statistically significant. Missing data were handled using only complete cases.

## Ethics statement

The National Centre for Health Statistics Research Ethics Review Board approved the protocols (ERB protocol numbers #2006–2007 and #2011–2017). The participants provided written informed consent.

## Results

### Characterization of the groups

The study group consisted of 11,579 individuals with a mean age of 45 years (range 20–79 years). In the whole population, 56% were never smokers, 22% ex-smokers and 22% current smokers. In the control group and asthma without FAO group, the prevalence of never smokers was almost 60%, while the asthma with FAO group had 26% never smokers and the FAO without asthma group had 24%. The individuals in the two FAO groups were older than controls and had been smoking more. In the FAO group without asthma, 5% had reported receiving a diagnosis of emphysema, 3% chronic bronchitis and 1% had been diagnosed with of cancer in the lung. The subject characteristics are further described in Table 1.

Anti-inflammatory medication was mainly used in the two asthma groups. The group with both asthma and FAO was prescribed more asthma medication (anti-inflammatory and or including bronchodilator therapy) (78% vs. 58%, p = 0.005) and more anti-inflammatory medication than asthmatics without FAO (43% vs. 26%, p = 0.002). They also more frequently reported presence of at least 2 asthma symptoms (65% vs. 49%, p = 0.004) and a longer

**Table 1. Characteristics of the asthma and FAO groups compared with controls.**

| Variable | Control n = 9,935 | Asthma without FAO n = 674 | p value Control-Asthma without FAO | Asthma with FAO n = 180 | p value Control-Asthma with FAO | FAO without asthma n = 790 | p value Control-FAO without asthma |
|---|---|---|---|---|---|---|---|
| Age* (mean ± SD) | 44.2 ± 15 | 43.4 ± 16 | 0.34 | 48.4 ± 15 | 0.002 | 51.2 ± 15 | <0.001 |
| Age > 44 years | 47% | 45% | 0.44 | 62% | 0.002 | 68% | < 0.001 |
| Female | 51% | 67% | < 0.001 | 59% | 0.14 | 47% | 0.018 |
| Hay fever | 16% | 39% | < 0.001 | 39% | <0.001 | 15% | 0.44 |
| **BMI group** | | | | | | | |
| < 25 | 31% | 25% | < 0.001 | 29% | 0.90 | 40% | < 0.001 |
| 25–30 | 34% | 29% | | 35% | | 35% | |
| ≥ 30 | 35% | 46% | | 36% | | 25% | |
| BMI (mean ± SD)* | 28.7 ± 6.6 | 30.8 ± 8.5 | <0.001 | 29.7 ±7.6 | 0.29 | 27.0 ± 6.0 | <0.001 |
| Pack-years, median (25th–75th percentile)† | 0 (0–4.2) | 0 (0–5) | 0.043 | 8 (0–27) | < 0.001 | 11.25 (0–30) | < 0.001 |
| Smoking history > 10 pack-years | 17% | 19% | 0.27 | 47% | < 0.001 | 50% | < 0.001 |
| **Smoking** | | | | | | | |
| Never smoker | 59% | 57% | 0.73 | 26% | < 0.001 | 24% | < 0.001 |
| Ex-smoker | 22% | 22% | | 30% | | 30% | |
| Current smoker | 20% | 21% | | 44% | | 46% | |
| Elevated cotinine | 25% | 25% | 0.94 | 49% | < 0.001 | 50% | < 0.001 |
| **Any of the below listed anti-inflammatory medication in the preceding month** | 1% | 26% | < 0.001 | 43% | < 0.001 | 4% | < 0.001 |
| None | 99% | 74% | < 0.001 | 57% | < 0.001 | 96% | < 0.001 |
| ICS ± LABA | 0% | 13% | | 31% | | 2.5% | |
| LTRA ± ICS ± LABA | 0% | 10% | | 8% | | 0% | |
| OCS ± ICS ± LTRA ± LABA | 1% | 3% | | 4% | | 1.5% | |

*Wald test,

†Wilcoxon rank sum test.

Abbreviations: B-Eos: blood eosinophils; B-Neu: blood neutrophils; BMI: body mass index; FAO: fixed airflow obstruction; FeNO: fraction of exhaled Nitric Oxide; ICS: inhaled corticosteroids; LABA: long acting beta agonist; LTRA: leukotriene receptor antagonist; OCS: oral corticosteroids

duration of asthma: median (25–75th percentile) 21 years (11–38 years) vs. 17 years (8–28 years, p = 0.001). No difference in age at asthma onset was seen: 51% in the asthma without FAO group had an onset of disease before 18 years of age vs. 57% in the asthma with FAO group, p = 0.30.

## Inflammatory patterns in the groups

The prevalence of elevated FeNO was 29% in the asthma without FAO group and 33% in the asthma with FAO group, compared with 17% among controls (p < 0.001 for both) (Fig 2A). Elevated FeNO was found in 16% of subjects with FAO without asthma, which was not significantly different from among control subjects (p = 0.59). Elevated B-Eos was more common in the asthma without FAO (30%), asthma with FAO (48%) and FAO without asthma (30%) groups, compared with the control group: 23% (p < 0.001, p < 0.001, and p = 0.003, respectively) (Fig 2B). Elevated B-Neu was also more prevalent in all case groups, asthma without FAO 29% (p = 0.028), asthma with FAO 35% (p = 0.013) and FAO without asthma 34% (p < 0.001), than among controls: 23% (Fig 2C). Elevated FeNO was more prevalent in both asthma groups than in the FAO without asthma group (Fig 2A) (p ≤ 0.001), and elevated

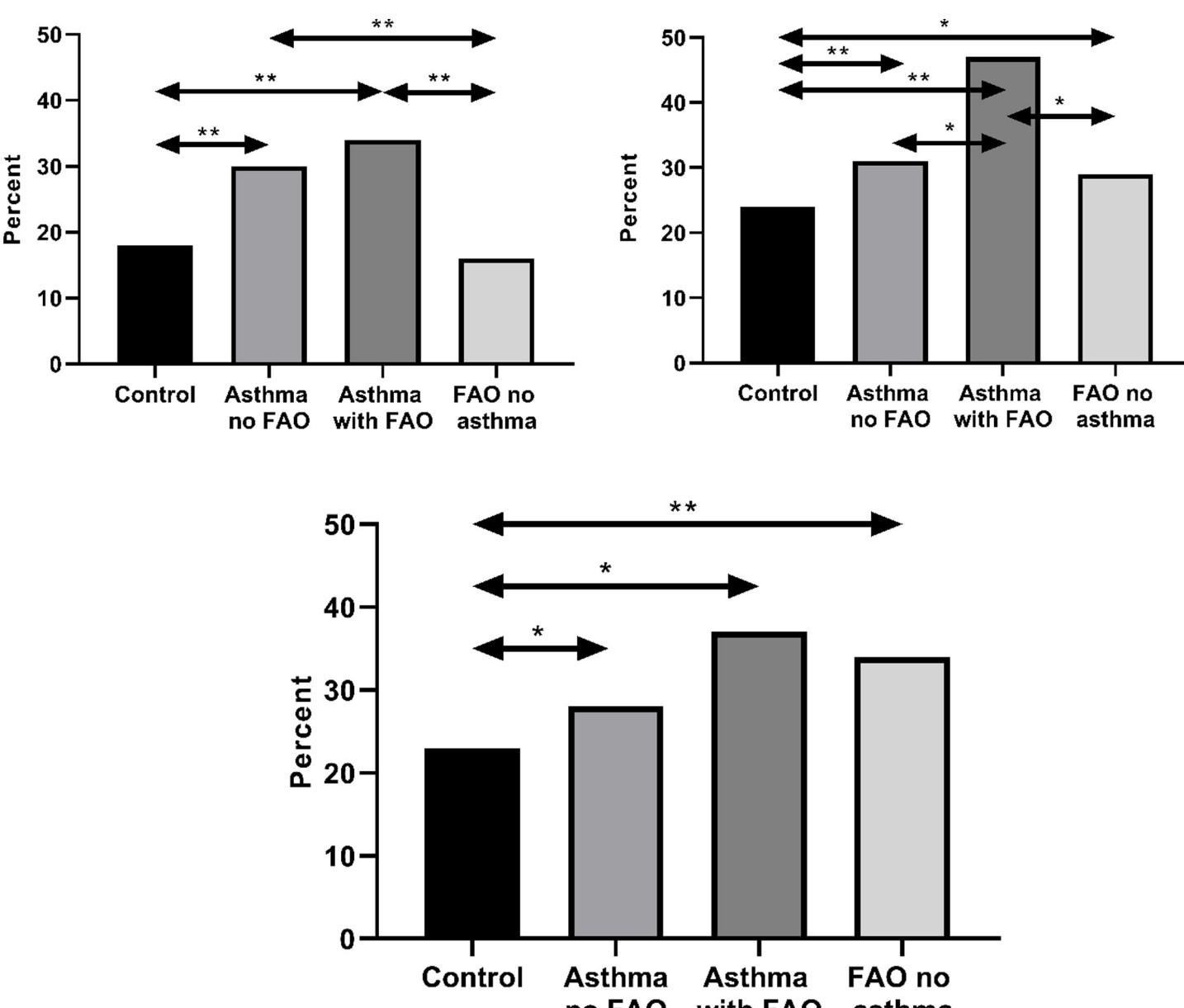

**Fig 2.** Percent with elevated marker in the groups respectively: Panel a: FeNO; panel b: B-Eos; panel c: B-Neu. (a) Percent with elevated FeNO ($\geq$25ppb) in the groups: Control: 17%, Asthma without FAO: 29%, Asthma with FAO: 33%, FAO without asthma: 16%, ** $p < 0.001$. (b) Percent with elevated B-Eos ($\geq$300 cells/μL) in the groups: Control: 23%, Asthma without FAO: 30%, Asthma with FAO: 48%, FAO without asthma: 30%, * $p < 0.05$, ** $p < 0.001$. (c) Percent with elevated B-Neu (B-Neu $\geq$5100 cells/μL) in the groups: Control: 23%, Asthma without FAO: 29%, Asthma with FAO: 35%, FAO without asthma: 34%, * $p < 0.05$, ** $p < 0.001$. Abbreviations: FeNO: fraction of exhaled Nitric Oxide, B-Eos: blood eosinophils, B-Neu: blood neutrophils, FAO: fixed airflow obstruction.

B-Eos was more common among the asthmatics with FAO than in the other groups ($p = 0.004$ versus asthma without FAO, and $p = 0.003$ versus FAO without asthma) (Fig 2B).

A higher prevalence of elevated FeNO was found in asthmatics compared with in controls among never-smokers. Also, those in the FAO group without asthma who were never smokers had an increased prevalence of elevated FeNO compared with controls (Table 2).

Among subjects with 10 or more pack-years' smoking, elevated B-Eos was significantly associated with asthma with FAO (Table 3).

**Table 2. Percent with elevated markers (FeNO ≥25ppb, B-Eos≥300 cells/μL, B-Neu ≥5100 cells/μL) in the asthma and FAO groups compared with the control group, stratified by smoking status.**

| Smoking status | Biomarker | Control | Asthma without FAO | p value control-asthma without FAO | Asthma with FAO | p value control-asthma with FAO | FAO without asthma | p value control-FAO without asthma |
|---|---|---|---|---|---|---|---|---|
| *Never smokers* | | n = 5,495 | n = 350 | | n = 53 | | n = 186 | |
| | Elevated FeNO | 19% | 33% | < 0.001 | 49% | 0.003 | 29% | 0.017 |
| | Elevated B-Eos | 20% | 24% | 0.006 | 38% | 0.12 | 27% | 0.11 |
| | Elevated B-Neu | 19% | 21% | 0.56 | 19% | 0.95 | 17% | 0.61 |
| *Ex-smokers* | | n = 2,077 | n = 147 | | n = 47 | | n = 225 | |
| | Elevated FeNO | 20% | 34% | 0.003 | 49% | 0.008 | 23% | 0.34 |
| | Elevated B-Eos | 25% | 37% | 0.014 | 61% | < 0.001 | 29% | 0.32 |
| | Elevated B-Neu | 21% | 23% | 0.64 | 24% | 0.78 | 23% | 0.59 |
| *Current smokers* | | n = 2,019 | n = 156 | | n = 76 | | n = 360 | |
| | Elevated FeNO | 7% | 15% | 0.017 | 11% | 0.16 | 5% | 0.19 |
| | Elevated B-Eos | 32% | 39% | 0.039 | 43% | 0.14 | 34% | 0.63 |
| | Elevated B-Neu | 38% | 57% | 0.002 | 54% | 0.068 | 49% | 0.003 |

Abbreviations: B-Eos: blood eosinophils; B-Neu: blood neutrophils; FAO: fixed airflow obstruction; FeNO: fraction of exhaled Nitric Oxide

**Table 3. Percent with elevated markers (FeNO ≥25ppb, B-Eos≥300 cells/μL, B-Neu ≥5100 cells/μL) in the asthma and FAO groups compared with controls, stratified by total smoke exposure (< 10 or ≥ 10 pack-years).**

| Smoke exposure | Biomarker | Control | Asthma without FAO | p value control-asthma without FAO | Asthma with FAO | p value control-asthma with FAO | FAO without asthma | p value control-FAO without asthma |
|---|---|---|---|---|---|---|---|---|
| *Smoking < 10 pack-years* | | n = 8,228 | n = 532 | | n = 107 | | n = 392 | |
| | **Elevated FeNO** | 18% | 33% | < 0.001 | 44% | < 0.001 | 21% | 0.17 |
| | **Elevated B-Eos** | 21% | 29% | < 0.001 | 44% | 0.004 | 26% | 0.15 |
| | **Elevated B-Neu** | 21% | 25% | 0.21 | 28% | 0.21 | 26% | 0.086 |
| *Smoking ≥ 10 pack-years* | | n = 1,707 | n = 142 | | n = 73 | | n = 398 | |
| | **Elevated FeNO** | 13% | 16% | 0.49 | 22% | 0.092 | 11% | 0.26 |
| | **Elevated B-Eos** | 32% | 35% | 0.43 | 53% | 0.025 | 34% | 0.60 |
| | **Elevated B-Neu** | 33% | 46% | 0.007 | 43% | 0.23 | 42% | 0.003 |

Abbreviations: B-Eos: blood eosinophils; B-Neu: blood neutrophils; FAO: fixed airflow obstruction; FeNO: fraction of exhaled Nitric Oxide

### Adjusted analysis

Elevated FeNO and elevated B-Eos were independently associated with asthma without FAO and asthma with FAO, and elevated B-Neu was related to FAO without asthma after adjusting for sex, ethnicity, BMI, age, steroid use in the preceding two days, study year, serum cotinine, smoking habits, and use of anti-inflammatory medication (Table 4).

### Interaction analyses for BMI and sex

The interaction analyses for BMI and sex showed significant interactions among asthmatics without FAO for BMI and elevated FeNO for the group with BMI < 25 (adjusted odds ratio (aOR): 4.0 (95% CI: 2.3–6.9)) compared with the group with BMI ≥ 30 (aOR: 2.3 (1.6–3.4); $p_{interaction}$ = 0.038). In the FAO without asthma group, there was an interaction with elevated B-Eos, more common at BMI < 25, (aOR: 1.5 (1.0–2.4)) than at BMI ≥ 30 (aOR: 0.74 (0.47–1.2); $p_{interaction}$ = 0.024). Female sex was significantly related to elevated FeNO in the FAO without asthma group aOR: 1.8 (1.2–2.8) as compared with male sex (aOR: 0.82 (0.60–1.1); $p_{interaction}$ = 0.003).

## Discussion

Asthma with FAO was, like asthma without FAO, associated with elevated levels of FeNO and B-Eos. However, asthmatics with FAO had the highest prevalence of elevated B-Eos of all groups. Elevated B-Neu was associated with FAO and smoking, both with and without asthma.

Both asthma with and without FAO were characterized by higher prevalence of elevated FeNO and B-Eos compared with controls. As asthma is an inflammatory disease, elevated levels of type-2 inflammatory markers are expected; we reported earlier that having increased levels of both biomarkers was related to higher prevalence of asthma morbidity, without accounting for presence of FAO [29]. These relations were consistent after adjusting for age and smoking, which had previously been reported in the literature to associate with affected B-Eos levels [30–33]. However, asthma with FAO was characterized by the highest prevalence of elevated B-Eos, a finding in line with our recent study in which we reported that increased levels of eosinophil activation markers were found in asthmatics with FAO [17]. This finding is of interest, as elevated B-Eos could signal a treatable component of the disease, and with more than half of the patients not using anti-inflammatory medication the preceding month, there might be further room for optimization of treatment.

Elevated B-Neu levels were found in non-asthmatic subjects with FAO. This is probably related to a large extent to smoking, known to be associated with increased B-Neu levels [12]

**Table 4. Adjusted\* odds ratios for elevated markers (FeNO ≥25ppb, B-Eos≥300 cells/μL, B-Neu ≥5100 cells/μL) for the asthma and FAO groups in relation to the control group in an adjusted logistic regression.**

| Biomarker | Asthma without FAO | P | Asthma with FAO | P | FAO without asthma | P |
|---|---|---|---|---|---|---|
| | a\*OR (CI) | | a\*OR (CI) | | a\*OR (CI) | |
| Elevated FeNO | 2.61 (2.09–6.97) | < 0.001 | 3.82 (2.02–3.37) | < 0.001 | 1.13 (0.86–1.48) | 0.37 |
| Elevated B-Eos | 1.38 (1.13–1.67) | 0.002 | 2.46 (1.43–4.22) | 0.002 | 1.25 (0.95–1.65) | 0.11 |
| Elevated B-Neu | 1.05 (0.82–1.35) | 0.81 | 1.23 (0.74–2.04) | 0.42 | 1.41 (1.09–1.86) | 0.010 |

\*Adjusted for sex, ethnicity, BMI, age, steroids in the preceding 2 days, study year, serum cotinine, smoking, anti-inflammatory medication in the preceding month.
Abbreviations: aOR: Adjusted odds ratio; B-Eos: blood eosinophils; B-Neu: blood neutrophils; BMI: body mass index; CI: 95% confidence interval; FAO: fixed airflow obstruction; FeNO: fraction of exhaled Nitric Oxide

and also that many in this group probably have COPD. Sub analyses performed in different smoking strata found this relation to be consistent only among current-smoking subjects. This was also found in the group with asthma without FAO, and with a trend in the asthma with FAO-group. Smoking in presence of respiratory or structural airway disease seems to relate to further increase of B-Neu. Elevated B-Neu could reflect decreased ability to resist chronic infections [34] in remodeled or fibrotic tissue, caused by smoking or injuries caused by other respiratory diseases.

The prevalence of elevated FeNO was low in the FAO without asthma group. As smoking is related to lower FeNO levels [35], we studied if this relation was consistent in different smoking strata. This finding appears to be driven mainly by smoking subjects as in the analysis limited to never smokers, elevated FeNO was more prevalent in the FAO without asthma group than among controls. Furthermore, we found a sex difference in subjects with FAO without asthma, where elevated FeNO associated with FAO without asthma among women, but not men. This could be explained by the same cut-off used in men and women despite differences reported in the literature, with lower levels in women than in men due to smaller airways [36]. BMI was also associated to different inflammatory pattern in asthmatics without FAO, with more active type-2 inflammation (FeNO and B-Eos) in subjects with normal BMI compared to obese subjects. Possible explanations for this finding could be both altered inflammation mechanisms in obese subjects with asthma, as higher proportion of neutrophilic asthma has been described in asthma with obesity [37–39]. Another explanation is that in obese subjects with asthma, respiratory symptoms might be more related to mechanical and ventilatory alterations [40, 41] rather than type 2 inflammation.

FeNO is a marker reflecting local type-2 inflammation in the airways, whereas both B-Eos and B-Neu are sampled in blood and do not necessarily mirror the inflammation in airway tissue [42]. However, while increased B-Eos levels have been linked to worse lung function [6, 7], the picture is more complicated when it comes to neutrophils. B-Neu is less correlated with the levels in other compartments, such as bronchoalveolar lavage, bronchial biopsies or sputum, than B-Eos [42, 43]. The increased amount of neutrophils in the airway lumen in COPD is well-established [44, 45], and a part of the group with FAO with or without asthma is likely to have COPD. Neutrophilia in airways and blood has, however, been associated with both better [46] and worse pulmonary function over time [47, 48]. Our findings rather link elevated B-Neu to airway disease (FAO, asthma) among smokers, as seen in the stratified analyses (Tables 2 and 3).

There are some limitations in this study. Lung function was only measured at one time point and there was no intervention with optimized treatment preceding the examinations. This makes the reliability of the FAO diagnosis weaker; non-reversibility can also be a symptom of under-treatment, though the asthma with FAO group reported using more medications. The cross-sectional study design does not allow for drawing conclusions on causality. Furthermore, the FAO without asthma group was heterogeneous even if it could be assumed that a significant proportion might be subjects with undiagnosed COPD and asthma. We know that COPD is still underdiagnosed [49] and the same might apply to asthma as the cases were defined based on self-reported physician-diagnosed asthma. In the light of this heterogeneity, this group rather represent the implications of fixed airflow obstruction than a common etiology for the impairment. B-Eos and B-Neu measured in blood do not necessarily reflect the levels in the airways [42, 43]. However, the advantage of their accessibility makes the clinical impact of these valuable to scrutinize.

In conclusion, in asthma with FAO the inflammatory pattern was characterized by eosinophilic inflammation and increased levels of FeNO, a pattern also seen in asthmatics without FAO. However, the eosinophil pattern was even more pronounced in asthmatics with FAO.

FAO in non-asthmatics was associated with elevated levels of B-Neu and our results suggested a close association to current smoking. This indicates a need for further studies to evaluate the role of both eosinophil and neutrophil inflammation in the development of fixed airflow obstruction.

## Supporting information

**S1 Questionnaires.**
(DOCX)

**S1 Dataset.**
(DOCX)

## Author Contributions

**Conceptualization:** Kjell Alving, João A. Fonseca, Christer Janson, Andrei Malinovschi.

**Data curation:** Ida Mogensen, Tiago Jacinto.

**Formal analysis:** Ida Mogensen, Tiago Jacinto.

**Funding acquisition:** Christer Janson, Andrei Malinovschi.

**Investigation:** Ida Mogensen.

**Methodology:** Andrei Malinovschi.

**Supervision:** Christer Janson, Andrei Malinovschi.

**Writing – original draft:** Ida Mogensen.

**Writing – review & editing:** Ida Mogensen, Tiago Jacinto, Kjell Alving, João A. Fonseca, Christer Janson, Andrei Malinovschi.

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
