## [Decision Letter · Decision Letter 0]

18 Sep 2020

PONE-D-20-25661

Inflammatory patterns in fixed airflow obstruction are dependent on the presence of asthma

PLOS ONE

Dear Dr. Mogensen,

Thank you for submitting your manuscript to PLOS ONE. After careful consideration, we feel that it has merit but does not fully meet PLOS ONE’s publication criteria as it currently stands. Therefore, we invite you to submit a revised version of the manuscript that addresses the points raised during the review process. Your manuscript loos interesting and provide some valauble data. some methodologial issues need to be resolved. Furhtermore please respond to the following comment: 

The group with FAO without asthma needs a more detailed approach. Which was the underlying airway disease?

  Please submit your revised manuscript by Nov 02 2020 11:59PM. If you will need more time than this to complete your revisions, please reply to this message or contact the journal office at plosone@plos.org. Please include the following items when submitting your revised manuscript:

We look forward to receiving your revised manuscript.

Kind regards,

Stelios Loukides

Academic Editor

PLOS ONE

Journal Requirements:

2. Please include the date(s) on which you accessed the databases or records to obtain the data used in your study.

3. Please include a copy of the interview guide used in the study, in both the original language and English, as Supporting Information, or include a citation if it has been published previously.

4.Thank you for stating the following financial disclosure:

 [The funders had no role in study design, data collection and analysis, decision to publish, or preparation of the manuscript.].

5. Please upload a new copy of Figure 1 as the detail is not clear. Please follow the link for more information: https://blogs.plos.org/plos/2019/06/looking-good-tips-for-creating-your-plos-figures-graphics/" https://blogs.plos.org/plos/2019/06/looking-good-tips-for-creating-your-plos-figures-graphics/" https://blogs.plos.org/plos/2019/06/looking-good-tips-for-creating-your-plos-figures-graphics/

Reviewers' comments:

Reviewer's Responses to Questions

**Comments to the Author**

1. Is the manuscript technically sound, and do the data support the conclusions?

Reviewer #1: Yes

2. Has the statistical analysis been performed appropriately and rigorously? 

Reviewer #1: I Don't Know

3. Have the authors made all data underlying the findings in their manuscript fully available?

Reviewer #1: Yes

4. Is the manuscript presented in an intelligible fashion and written in standard English?

Reviewer #1: Yes

5. Review Comments to the Author

Reviewer #1: To the authors

General Comments:

This is an interesting study investigating the relation of inflammatory markers to fixed airflow obstruction depending on the presence of asthma. The results of the study have been previously reported in the literature, however fixed airway obstruction in asthmatics and non-asthmatics remains a clinical entity which requires further research. The main strength of the study is the large population that was used. There are a few suggestions and questions regarding mainly the classification of patients into the four clinical groups and the heterogeneity of the FAO without asthma group. Overall, it is a well-written and well-organized article with an interesting hypothesis and a thorough discussion.

Specific Suggestions/Questions:

1. Materials and methods section, “spirometry, questionnaires and body measures” paragraph:

a. “For the subjects with a non-obstructive pattern (FEV1/FVC ≥ 0.7 and above LLN), only pre bronchodilatory values were collected”. It is known that there are patients with asthma who present non-obstructive FEV1/FVC pre- bronchodilation values. Diagnosis of the disease will require a history of symptoms suggestive of asthma, a pre and post bronchodilation test or even a bronchial challenge test. Consequently, it is possible that some patients classified as healthy controls may have asthma and this should be mentioned in the discussion section.

b. “Asthma diagnosis was self-reported, but should be diagnosed by a health professional”. Please clarify if all patients reported as asthmatics in the study were officially diagnosed by an experienced health care professional. A high percentage of patients were not receiving any kind of treatment, suggesting that most of them may had not been diagnosed and managed by any experienced physician. Therefore, as mentioned by the authors, the fixed airflow obstruction observed in some asthmatics possibly reflects the lack of optimal treatment. The fact that one-time point spirometry data were available for each patient is another caveat in the characterization of subjects as asthmatics or controls, which is also recognized by the authors.

2. Results section, “characterization of the groups” paragraph:

“Anti-inflammatory medication was mainly used in the two asthma groups. The group with both asthma and FAO was prescribed more asthma medication (78% vs. 58%, p = 0.005)”. It is shown in table 1 that 57% of patients with asthma and FAO and 74% of patients with asthma and no FAO were not receiving any medication. Please clarify this point.

3. Results section, “Interaction analyses for BMI and sex” paragraph:

“The interaction analyses for BMI and sex showed significant interactions among asthmatics without FAO for BMI and elevated FeNO for the group with BMI < 25 (adjusted odds ratio (aOR): 4.0 (95% CI: 2.3–6.9)) compared with the group with BMI ≥ 30 (aOR: 2.3 (1.6–3.4);pinteraction = 0.038). In the FAO without asthma group, there was an interaction with elevated B-Eos, more common at BMI < 25, (aOR: 1.5 (1.0–2.4)) than at BMI ≥ 30 (aOR: 0.74 (0.47–240 1.2); pinteraction = 0.024)”. It would be interesting if some hypothesis explaining these interactions could be added in the discussion.

4. Discussion section: The discussion is well structured and analyzes thoroughly the results on the basis of current literature data, with the exception of the results mentioned in comment number 4. The limitations of the study are also well recognized by the authors. Besides possible mis-diagnosis of subjects as healthy controls or asthmatics, based on comment 2a and 2b, characterization of the FAO without asthma group is also problematic. This cluster of patients seems to be significantly heterogeneous and probably involves subjects with various conditions such as COPD, asthma - COPD overlap, underdiagnosed asthma not receiving treatment, occupational diseases and fixed airflow obstruction due to any inhaled irritating factor. Consequently, the underlying inflammatory pattern seems also unclear to my opinion. It is mentioned that FAO without asthma group presented statistically significant higher levels of B-Eos and B-Neu compared to healthy controls and this possibly reflects an expected underlying inflammatory process that led to fixed obstruction [lines 175-180: “Elevated B-Eos was more common in the asthma without FAO (30%), asthma with FAO (48%) and FAO without asthma (30%) groups, compared with the control group: 23% (p < 0.001, p < 0.001, and p = 0.003, respectively) (Fig 2b). Elevated B-Neu was also more prevalent in all case groups, asthma without FAO 29% (p = 0.028), asthma with FAO 35% (p = 0.013) and FAO without asthma 34% (p < 0.001), than among controls: 23% (Fig 2c)”]. The stratified by smoking status analysis of the inflammatory markers (table 2) shows no significant increase of B-Eos in any of the three strata of the FAO without asthma group, whereas the only sub-group of these patients that presented significantly higher B-Neu were current smokers. Finally, the adjusted analysis (table 4) showed that increased B-Neu levels were independently associated to FAO without asthma. This latter association, as recognized by the authors, is most probably due to smoking habits.

Would it be possible to make a further, etiology-based, sub-classification of patients with FAO without asthma so as to make more accurate observations on the underlying inflammatory pattern?

Given the great heterogeneity of the FAO without asthma group, is it safe to draw any conclusions on the inflammatory pattern for these patients in a uniform manner, such as the correlation with elevated B-Neu? If so, are there any clinical implications of this finding?

5. References: Please add the following reference in relation to lines 57-59 in the introduction: Konstantellou et al. Persistent airflow obstruction in patients with asthma: Characteristics of a distinct clinical phenotype. Respir Med 2015 Nov;109(11):1404-9.

6. PLOS authors have the option to publish the peer review history of their article (what does this mean?). If published, this will include your full peer review and any attached files.

Reviewer #1: No

---

## [Author Response · Author response to Decision Letter 0]

30 Oct 2020

Reply to the reviewers:

Reviewer #1: To the authors

General Comments:

This is an interesting study investigating the relation of inflammatory markers to fixed airflow obstruction depending on the presence of asthma. The results of the study have been previously reported in the literature, however fixed airway obstruction in asthmatics and non-asthmatics remains a clinical entity which requires further research. The main strength of the study is the large population that was used. There are a few suggestions and questions regarding mainly the classification of patients into the four clinical groups and the heterogeneity of the FAO without asthma group. Overall, it is a well-written and well-organized article with an interesting hypothesis and a thorough discussion.

Specific Suggestions/Questions:

1. Materials and methods section, “spirometry, questionnaires and body measures” paragraph:

a. “For the subjects with a non-obstructive pattern (FEV1/FVC ≥ 0.7 and above LLN), only pre bronchodilatory values were collected”. It is known that there are patients with asthma who present non-obstructive FEV1/FVC pre- bronchodilation values. Diagnosis of the disease will require a history of symptoms suggestive of asthma, a pre and post bronchodilation test or even a bronchial challenge test. Consequently, it is possible that some patients classified as healthy controls may have asthma and this should be mentioned in the discussion section.

Thank you for this comment, and yes indeed, a significant reversibility could be missed among them with only prebronchodilatory measurements. However, the label asthma was based on a reported asthma diagnosis received by a health professional and not on the spirometry findings. This is clarified in the Methods (page 5, rows 101-102) and discussed in Discussion (page 17, rows 299-300).

b. “Asthma diagnosis was self-reported, but should be diagnosed by a health professional”. Please clarify if all patients reported as asthmatics in the study were officially diagnosed by an experienced health care professional. A high percentage of patients were not receiving any kind of treatment, suggesting that most of them may had not been diagnosed and managed by any experienced physician. Therefore, as mentioned by the authors, the fixed airflow obstruction observed in some asthmatics possibly reflects the lack of optimal treatment. The fact that one-time point spirometry data were available for each patient is another caveat in the characterization of subjects as asthmatics or controls, which is also recognized by the authors.

We have rewritten for better clarity this in the Methods part (page 5 rows 103-105).

2. Results section, “characterization of the groups” paragraph:

“Anti-inflammatory medication was mainly used in the two asthma groups. The group with both asthma and FAO was prescribed more asthma medication (78% vs. 58%, p = 0.005)”. It is shown in table 1 that 57% of patients with asthma and FAO and 74% of patients with asthma and no FAO were not receiving any medication. Please clarify this point.

Thank you for pointing this ambiguity out. “Asthma medication” is referring to any asthma medication (LABA and SABA included), while anti-inflammatory medication (inhaled corticosteroids, leukotriene receptor antagonists and oral corticosteroids) are presented separately in Table 1. This is now more clearly specified in the text now. Results section: Page 10 row 168-169

3. Results section, “Interaction analyses for BMI and sex” paragraph:

“The interaction analyses for BMI and sex showed significant interactions among asthmatics without FAO for BMI and elevated FeNO for the group with BMI < 25 (adjusted odds ratio (aOR): 4.0 (95% CI: 2.3–6.9)) compared with the group with BMI ≥ 30 (aOR: 2.3 (1.6–3.4);pinteraction = 0.038). In the FAO without asthma group, there was an interaction with elevated B-Eos, more common at BMI < 25, (aOR: 1.5 (1.0–2.4)) than at BMI ≥ 30 (aOR: 0.74 (0.47–240 1.2); pinteraction = 0.024)”. It would be interesting if some hypothesis explaining these interactions could be added in the discussion.

This findings are discussed in Discussion Page 17 rows row 283-289

4. Discussion section: The discussion is well structured and analyzes thoroughly the results on the basis of current literature data, with the exception of the results mentioned in comment number 4. The limitations of the study are also well recognized by the authors. Besides possible mis-diagnosis of subjects as healthy controls or asthmatics, based on comment 2a and 2b, characterization of the FAO without asthma group is also problematic. This cluster of patients seems to be significantly heterogeneous and probably involves subjects with various conditions such as COPD, asthma - COPD overlap, underdiagnosed asthma not receiving treatment, occupational diseases and fixed airflow obstruction due to any inhaled irritating factor. Consequently, the underlying inflammatory pattern seems also unclear to my opinion. It is mentioned that FAO without asthma group presented statistically significant higher levels of B-Eos and B-Neu compared to healthy controls and this possibly reflects an expected underlying inflammatory process that led to fixed obstruction [lines 175-180: “Elevated B-Eos was more common in the asthma without FAO (30%), asthma with FAO (48%) and FAO without asthma (30%) groups, compared with the control group: 23% (p < 0.001, p < 0.001, and p = 0.003, respectively) (Fig 2b). Elevated B-Neu was also more prevalent in all case groups, asthma without FAO 29% (p = 0.028), asthma with FAO 35% (p = 0.013) and FAO without asthma 34% (p < 0.001), than among controls: 23% (Fig 2c)”]. The stratified by smoking status analysis of the inflammatory markers (table 2) shows no significant increase of B-Eos in any of the three strata of the FAO without asthma group, whereas the only sub-group of these patients that presented significantly higher B-Neu were current smokers. Finally, the adjusted analysis (table 4) showed that increased B-Neu levels were independently associated to FAO without asthma. This latter association, as recognized by the authors, is most probably due to smoking habits.

Would it be possible to make a further, etiology-based, sub-classification of patients with FAO without asthma so as to make more accurate observations on the underlying inflammatory pattern?

Given the great heterogeneity of the FAO without asthma group, is it safe to draw any conclusions on the inflammatory pattern for these patients in a uniform manner, such as the correlation with elevated B-Neu? If so, are there any clinical implications of this finding?

Thank you for this comment. It is indeed a crucial question how to interpret these results. We have now made some modifications in the abstract, results section and discussion: 

The group with FAO without asthma is indeed heterogeneous, but the definition of this group does not imply a common etiology but is a sole common characteristic. This in now further discussed in the limitations section of the discussion. The available additional respiratory diagnoses are added in the results section (they could only explain about one tenth of the cases). We discuss also that a group with un(der)diagnosed COPD, as well as undiagnosed asthma, might be the most plausible etiologies found in this group. Also the main conclusion is to some extent modified.

Abstract, Results: Page 9, rows 156-158, Discussion: Page 16 Row 252-253, page 18 rows 273-274, page 17 row 306-3011

5. References: Please add the following reference in relation to lines 57-59 in the introduction: Konstantellou et al. Persistent airflow obstruction in patients with asthma: Characteristics of a distinct clinical phenotype. Respir Med 2015 Nov;109(11):1404-9.

Thank you for this suggestion, the reference is now added Page 3 row 61

---

## [Decision Letter · Decision Letter 1]

16 Nov 2020

Inflammatory patterns in fixed airflow obstruction are dependent on the presence of asthma

PONE-D-20-25661R1

Dear Dr. Mogensen,

We’re pleased to inform you that your manuscript has been judged scientifically suitable for publication and will be formally accepted for publication once it meets all outstanding technical requirements.

Kind regards,

Stelios Loukides

Academic Editor

PLOS ONE

Additional Editor Comments (optional):

Reviewers' comments:

Reviewer's Responses to Questions

**Comments to the Author**

1. If the authors have adequately addressed your comments raised in a previous round of review and you feel that this manuscript is now acceptable for publication, you may indicate that here to bypass the “Comments to the Author” section, enter your conflict of interest statement in the “Confidential to Editor” section, and submit your "Accept" recommendation.

Reviewer #1: All comments have been addressed

2. Is the manuscript technically sound, and do the data support the conclusions?

Reviewer #1: Yes

3. Has the statistical analysis been performed appropriately and rigorously? 

Reviewer #1: Yes

4. Have the authors made all data underlying the findings in their manuscript fully available?

Reviewer #1: Yes

5. Is the manuscript presented in an intelligible fashion and written in standard English?

Reviewer #1: Yes

6. Review Comments to the Author

Reviewer #1: To the Author

Thank you for your quality revision of this manuscript. All comments and suggestions are addressed appropriately. For these reasons, I would recommend acceptance of this article.

Kind regards

7. PLOS authors have the option to publish the peer review history of their article (what does this mean?). If published, this will include your full peer review and any attached files.

Reviewer #1: No

---

## [Editor Report · Acceptance letter]

23 Nov 2020

PONE-D-20-25661R1 

Inflammatory patterns in fixed airflow obstruction are dependent on the presence of asthma 

Dear Dr. Mogensen:

I'm pleased to inform you that your manuscript has been deemed suitable for publication in PLOS ONE. Congratulations! Your manuscript is now with our production department. 

Kind regards, 

on behalf of

Dr Stelios Loukides 

Academic Editor

PLOS ONE